# Use of tobacco during COVID-19: A qualitative study among medically underserved individuals

Tamar Klaiman[1]*, Nsenga Farrell[1], Dorothy Sheu[1], Aerielle Belk[1], Jasmine Silvestri[1], Jannie Kim[2], Ryan Coffman[3], Joanna Hart[1]

1 Palliative and Advanced Illness Research Center, University of Pennsylvania, Philadelphia, PA, United States of America, 2 CHDI Foundation, New York, NY, United States of America, 3 Tobacco Policy and Control Program, Philadelphia Department of Public Health, Philadelphia, PA, United States of America

* tamar.klaiman@pennmedicine.upenn.edu

**Data Availability Statement:** The datasets generated and analyzed during the current study are not publicly available because informed consent stated that any data collected would not be

## Abstract

The COVID-19 pandemic produced stress for people around the world. The perception that tobacco can be a coping tool for stress relief suggests that the conditions during the COVID-19 pandemic can provide insight into the relationship between stress and tobacco use patterns, particularly among those most at risk for severe COVID-19 disease. The goal was to identify the impacts of the COVID-19 pandemic on tobacco use and preparedness for smoking cessation among individuals who smoke and are older and medically underserved. We conducted in-depth interviews with 39 patients to learn about individuals' smoking behavior during the COVID-19 pandemic. We used a modified grounded theory approach to code and analyze all qualitative data. We conducted thematic analysis to identify key factors associated with smoking behaviors during COVID-19. Our results indicated that increases in perceived stress and social isolation may have been associated with increased tobacco use during the COVID-19 pandemic. Pandemic-related social isolation contributed to increases in smoking, despite respondents being concerned about the severity of COVID-19. While many respondents felt that smoking relieved their stress from the pandemic, they appeared unaware of the stress-inducing properties of tobacco use. Our findings indicate that pandemic-related stress impacted smoking behavior among older, medically underserved smokers. Results may assist clinicians in addressing the role of tobacco use in response to highly stressful events. Smoking cessation strategies should consider the implications of stress on smoking behavior, including smoking relapse in response to highly stressful events–particularly for medically underserved populations.

## Introduction

The COVID-19 pandemic produced exceptional amounts of prolonged financial, social, physical, and emotional stress on the global community [1]. The persistent, but incorrect, perception that tobacco can be a coping tool for stress relief [2–4] suggests that the conditions during

used in future studies; however, de-identified data may be available from the PAIR Center Data Manager at pair@pennmedicine.upenn.edu on reasonable request.

**Funding:** This project was funded by the Patient Centered Outcomes Research Institute (PCORI) (PCORI ID: PCS-2018C1-11326 (JH)). The funders had no role in study design, data collection and analysis, decision to publish, or preparation of the manuscript.

**Competing interests:** No authors have competing interests.

the COVID-19 pandemic can uniquely provide important insight into the relationship between complex stress and tobacco use patterns.

There is minimal existing knowledge of how patients navigated and managed contextual forces, including stress, that may have influenced their tobacco use behaviors and attitudes and led to variations in use patterns during the pandemic [5–7]. Existing evidence of tobacco use during the COVID-19 pandemic suggests that there was variability in whether individuals increased or decreased their use, without a consistent pattern [8]. For example, survey-based studies investigating tobacco use during the COVID-19 pandemic have consistently revealed that approximately a quarter of individuals increased tobacco use, approximately a quarter decreased tobacco use, and approximately half experienced no change in their use [9–12] Yet, the evidence to date is limited and does not sufficiently explore the mechanisms or causes underlying these patterns [1, 13–15] which are important for leveraging the knowledge gained to improve tobacco treatment efforts. The social distancing restrictions enacted during COVID-19 changed and may have restricted access to health care, including tobacco treatment, but may have also reduced or changed access to tobacco products.

Further, while there is evidence that age and perceived risk of severe COVID-19 disease was associated with tobacco use patterns, there has been little focused attention to the tobacco use patterns of those most at risk for severe COVID-19 disease [5, 16, 17]. Individuals who are low income and historically marginalized were at especially high risk for severe COVID-19 disease, unemployment, and reduced access to health care [18, 19]. These same populations are also disproportionately impacted by tobacco use resulting from targeted tobacco marketing strategies, inequitable access to tobacco treatment, and chronic stress [20]. For example, tobacco companies market tobacco to low-income populations and people of color [21], distribute cigarettes to children in low income neighborhoods [22], support more retailers and more retail window advertising in low income communities [23], discount prices to improve uptake [24], and have increased the nicotine concentration of cigarettes over time [25, 26]. Prior research has also identified contextual factors such as social norms, economic structures, and high levels of stress from racial discrimination and poverty as contributing to tobacco use among Black and low-income groups [27]. The pervasive structural stress among these medically underserved groups was further amplified during the COVID-19 pandemic. Therefore, the primary goal of this project was to identify the impacts of the COVID-19 pandemic on tobacco use and preparedness for smoking cessation among individuals who smoke and who are older and medically underserved. Utilizing in-depth qualitative interviews allows contextualization around the patterns seen in the quantitative data that has been published.

## Methods

We conducted a qualitative study consisting of in-depth interviews with older, medically underserved adults to learn about their smoking behavior during the COVID-19 pandemic. We recruited participants from September 30, 2020, to September 14, 2021 –during the height of the pandemic through the initiation of vaccines. Patient participants were recruited through multiple sampling strategies. First, we utilized the electronic health record of the University of Pennsylvania Health System (UPHS) to identify potential participants. We randomly selected UPHS outpatients who were at least 50 years old and had electronic health record (EHR)-documented daily tobacco use. We sent batches of mailed recruitment letters in randomized fashion to 25 to 50 potentially eligible patients at a time asking them to complete an online screening survey to determine eligibility. Using data available in the EHR, we purposefully sampled for rurality based on ZIP code (i.e., a Rural-Urban Community Area score of $\geq$2) during later periods of recruitment (i.e., subsequent batches of letters) given that we lacked

sufficient rural representation in our initial group of participants and to ensure diverse representation of perspectives.

Second, we approached potential participants through our established partnership with a local, Hispanic serving non-profit agency. Trained staff—as part of the organization's COVID-19 Mobile Response Unit outreach to low-income urban and rural areas in northeastern and central Pennsylvania—recruited and assessed for eligibility among individuals engaging in their outreach sessions. Third, we relied on snowball sampling, in which participating patients provided information about the study to acquaintances who were then assessed for eligibility using the same approach.

After providing verbal informed consent for participation and documenting it in the recruitment spreadsheet, enrolled individuals participated in a structured telephone interview that was audio recorded with their permission. Consents were collected verbally because all interviews were conducted via telephone. The University of Pennsylvania IRB approved all procedures, including a verbal telephone consent procedure. Participants were given a full description of the study and agreed to participate after expressing understanding of the study procedures. Eligible individuals were using tobacco daily, 50 to 80 years old, and represented at least one of the following demographic groups: reported household income less than 200% of the federal poverty line based on household size, less than a high school level of formal educational attainment, self-identified as Black or African American, Hispanic or Latinx, and/or reported living in a rural area. Only those who spoke English or Spanish were included. Patients received $50 USD for their participation. Interviews were conducted in English or Spanish based on participants' preferences.

The interview guide (Supplement 1) was developed through literature reviews, in consultation with clinicians who treat lung disease and/or tobacco use disorder, and a stakeholder advisory committee consisting of patient and community advocacy groups, patients, clinicians, and health system leaders. The interviews focused on (1) smoking behavior prior to and during the COVID-19 pandemic, (2) perceptions about the impact of smoking on the development and severity of COVID-19, (3) barriers and facilitators for tobacco treatment prior to and during the COVID-19 pandemic, and (4) stressors and lifestyle changes during the COVID-19 pandemic. All questions were open ended and reflected high-level concepts (such as motivation, or behavior), and all respondents were asked questions focused on these core topics. Recruitment continued until thematic saturation was reached–no new themes arose after three additional interviews [28]. A professional transcription and translation company transcribed the audio recordings of the interviews and, if needed, translated the text to English from Spanish. The transcription process also removed identifying information in preparation for analysis. The Institutional Review Board of the University of Pennsylvania approved all procedures (protocol #843833).

## Analysis

We used a modified grounded theory approach to code and analyze all qualitative data. Investigators inductively developed a preliminary codebook based on emerging themes and deductively based on behavioral health constructs identified in the literature review (JH), including beliefs about consequences, motivations and goals, environmental context, and resources. All coding was overseen by a Senior Qualitative Research Scientist (TK). Research staff trained in qualitative methods conducted the coding. The codebook was iteratively revised as additional transcripts were reviewed. Research staff used the codebooks to code all transcripts in NVIVO 12.0, with 20% coded by two staff members to assess reliability. Research staff met at least weekly during coding to resolve conflicts. Any disagreements were discussed until consensus

was reached with iterative revision of the codebook and review of previously coded transcripts as necessary. Theme sheets were developed for each of the identified themes, and weekly meetings were held with study staff to analyze results within and between themes. The University of Pennsylvania IRB approved this study (protocol #843833).

## Results

Thirty-nine patients completed interviews. Respondent demographics are noted in Table 1.

Unrelated to the pandemic, most respondents had attempted to quit tobacco at least once, yet all resumed smoking. Participants identified triggers such as increased anxiety, stress, grief, other substances (e.g., coffee or alcohol), and general boredom as precipitating smoking relapse even after prolonged abstinence.

> *I stopped smoking–I forget when it was, I think it was in May. I was doing good, and I think my brother-in-law passed in September. . .and he was my favorite brother-in-law, and when he passed away, it made me feel so bad. So, that's how I started back smoking. (Respondent 115)*

> *Like if I drink coffee, like every time I drink coffee throughout the day. (Respondent 033)*

> *Well, I went out drinking and. . . someone was smoking and you know, I was like, "Oh, let me get one." They were like, "Oh no, you don't smoke anymore?" And I was like, "No, give me one." And that's what happened. (Respondent 106)*

Tobacco was frequently used by participants as a tool for stress relief. Respondents felt that smoking helped them to calm down, particularly after stressful events or managing grief and anger. The reliance on tobacco as a coping tool for stress was both a precipitant of relapse and a reason to continue use. Some framed their tobacco use for this purpose as distinct from more general dependence on nicotine or smoking.

**Table 1. Respondent demographics.**

| Respondents N = 39 (%) | |
|---|---|
| **Age** | |
| 50–54 | 6 (15.4) |
| 55–59 | 7 (17.9) |
| 60–64 | 13 (33.3) |
| 65–69 | 10 (25.6) |
| 70–74 | 2 (5.1) |
| 75+ | 1 (2.6) |
| **Gender** | |
| Man | 20 (51.3) |
| Woman | 19 (48.7) |
| **Race** | |
| White or Caucasian | 13 (33.3) |
| Black or African American | 19 (48.7) |
| Asian or Asian American | 0 (0.0) |
| Native American | 0 (0.0) |
| Other | 7 (17.9) |

*You know, like. . . I don't know, many people don't understand it, but it really does calm you down. Well, some people need it because they're regular smokers, but in my case, I see it as a relaxant, for me. (Respondent 022)*

*. . . I did good for six months, and then one day I got angry, and I bought a packet of cigarettes and did myself in. (Respondent 107)*

*I once stopped smoking, and I stopped for. . . I'm not lying, for around 8 or 9 months, but when my mom passed away, when I lost my mom, I went back to smoking. I haven't quit ever since. (Respondent 024)*

The impact of COVID-19 on reporting smoking behavior was varied among respondents. Potential pandemic-era structural barriers to accessing tobacco products, such as stay-at-home orders and social distancing requirements, did not play a major role in tobacco purchase or use patterns among our respondents. For example, almost all respondents did not have trouble buying cigarettes from local convenience stores using their same purchasing behaviors. However, participants did report following recommended COVID-19 mitigation measures to reduce their risk of contracting the virus, such as hand washing, mask wearing, and avoiding crowds.

*I go to the store and I buy cigarettes by the carton. So, that's the same way I do now. (Respondent 217)*

*I could be out of cigarettes, and make it for a couple of hours, and then it wouldn't matter what time of the day or night it was. [laughs] If I was still awake, I'd go out and get a pack. So, I basically purchase them the say way, except now I have to put a mask on when I got into the store. (Respondent 158)*

*Well, it basically hasn't changed. . . we have what they call, "Cigarette Outlet Stores." . . . They sell cigarettes. . . Just one particular one that I know is kind of . . . tucked away, and it's never very crowded when I go there. So, it's—it's nice in that respect because you're not around a whole lot of other people, because the store is at least—well, see that's another change in my routine too is, I stay away from the stores on Fridays, Saturdays, and Sundays. I will prefer to go usually on a Tuesday, if I can, because I find that Tuesday for some reason, just in this general area where I live, that it's not as crowded on Tuesday as it is any other day. (Respondent 156)*

Some respondents reported increasing their tobacco use directly due to pandemic-related changes. These included increased anxiety, boredom, or having more unfilled or leisure time due to COVID-related activity restrictions and social distancing. The primary source of anxiety identified by respondents was an increase in social isolation due to the pandemic.

*. . . I've always kind of like had a problem with smoking. . . I can say, oh, it wasn't as bad as it is now. You know. . . being . . .in the house, and you kind of feel like closed in. And so, it's kind of like. . . it was kind of like comforting for me. So, I think I'm smoking more. . . than before. (Respondent 033)*

*I would say like on average, like when I'm home more due to the pandemic, I've been smoking more. (Respondent 141)*

*I find myself—it seems that I'm smoking more because of [the COVID-19 pandemic], with the stress and being in the house, and not being able to do things, and being around people. (Respondent 132)*

*Either, either boredom, which is heightened during, um COVID, books or boredom. Or maybe frustration, you know, maybe frustration that I'm not doing very much, which I guess is kind of like boredom. (Respondent 120)*

Many respondents believed that, if contracted, their severity of COVID-19 illness would be worse due to tobacco use or due to co-morbidities also tied to tobacco use. This caused increased stress and anxiety among participants who worried about their susceptibility to COVID-19 and the immediate health threat of the virus, including heightened attention to possible mortality. This perception was based on their recognition that tobacco is a pulmonary toxin and COVID-19 is a respiratory virus.

*I have a cigarette addiction, but I also need to know that I'm not a millionaire and I can't spoil myself as I used to. I used to not care because, I was able to touch every door and touch any-thing, now you can't touch everything and you need to be careful of someone touching you from behind because... It's gotten me a bit more anxious, because I start thinking, "what if this virus can't be cured or something?" You know? So one just gets sort of scared. (Respondent 022)*

*And people who smoke are more susceptible to catch that thing, because your lungs aren't strong enough to fight off that disease, because people who smoke, carry nicotine in their lungs and that makes it harder to breathe. (Respondent 024)*

*I've figured that at some point in the early—early stages of the pandemic, when it was first being announced, I smoke more, and then I was panicky. I was a little nervous about what was going on. I was nervous at how susceptible I would be to it... I think COVID would kill me. I've already got a weakened immune system, and then the things that do I hear about it, you know, I have secondary hypertension, diabetes, um, nephrotic syndrome. (Respondent 044)*

In part due to these health risks, some respondents described an initial decrease in their tobacco use during the early spread and awareness of COVID-19. However, often related to an acute stressor resulting from the pandemic conditions, respondents described relapses.

*Well, from June [2020] I went back down again. When me and my doctor, we first started I think it was in September that we started. And, like in October because I'm taking these pills of Bupropion, and they were helping like a little bit. One point, I was down to like, like four cigarettes a day. And then maybe end of October, I had, I had tragedy in my family. I picked up smoking a little more after that. (Respondent 110)*

*I'd stopped like somewhere in May and I was doing good. And then when I got the call about my brother in-law, and that made me go out straight to the store and buy a pack of cigarette. Now, that's the time I was smoking cigarette one after another because I couldn't get over, you know, what happened to my brother in-law and that's how it all started. (Respondent 115)*

*And then we moved here and I started smoking again, because I was so worried. My nerves are so bad that I was worried about us paying the rent because we're paying that like almost double of the rent we were paying before. (Respondent 122)*

## Discussion

Our results indicate that, for many respondents, increases in perceived stress and social isolation may have been associated with increased tobacco use during the COVID-19 pandemic. While smoking is associated with increased severity of COVID-19 symptoms [1, 2, 4] and

worse outcomes [29], respondents did not appear to rely on the emerging scientific evidence to inform this opinion. In contrast, their perceptions of illness and resulting personal health risks were based on personal experience. While many respondents felt that smoking relieved their stress from the pandemic, they appeared unaware of the stress-inducing properties of tobacco use [4]. Therefore, a possible explanation for increased tobacco use, despite this increased risk perception, is that many individuals relied on tobacco as a maladaptive coping tool in the face of acute stress. Because acute stress was increased during the pandemic, this may have further exacerbated tobacco use patterns and may have prevented quit attempts despite the "teachable moment" of tobacco-associated COVID-19 risk.

In other settings, near-term health threats have been effective in promoting tobacco treatment uptake and smoking cessation. For example, lung cancer screening is applicable to patients with many years of tobacco use and as many as half continue to use tobacco at the time of screening [30]. Tobacco treatments are successful in the context of lung cancer screening because of the inherent health and mortality threat patients confront when undergoing the annual screening which is explicitly performed to detect life-threatening cancer [31, 32]. Therefore, a health threat (risk of cancer) becomes a strategic moment for patients who may approach tobacco cessation as carrying immediate health benefits. However, our results suggest that tobacco treatment efforts in the context of health-related stress may be augmented by including specific stress management tools. In the case of COVID-19, which carried not just health threats but also immediate financial and social threats, integrating stress management interventions into tobacco treatment during this critical moment may have resulted in fewer patients increasing tobacco use in response.

The strengths of our study include the representation of patients from medically underserved groups and our use of qualitative methods to gain greater insight into the observed patterns of tobacco use during the pandemic [5, 7, 16]. Our study has limitations in addition to its strengths. First, our participants were recruited from a single state (Pennsylvania). This is particularly important given state-specific COVID-19 policies and burdens during the period of study. However, Pennsylvania is a geographically and politically diverse state, and we developed recruitment processes to maximize the sample's diversity. Second, we conducted cross-sectional interviews, such that we were unable to describe longitudinal patterns or changes in individuals' perspectives. The study was purely observational, which limits our ability to assess causal mechanisms for increased tobacco use. COVID-related research restrictions included remote-only recruitment, which led to challenges in response rate as we relied on EHR documentation of contact information and mail and phone contacts only. However, such approaches also may have engaged patients who attended in-person appointments less regularly given that we relied on recruitment based on prior visits (i.e., panels of patients) rather than ongoing engagement with health care services. Future work in this area should consider utilizing the concepts we identified as salient to develop a large-scale survey that would allow for statistical comparisons and analyses.

## Conclusion

Perceived stress is a well-described risk factor for tobacco smoking and a barrier to successful smoking cessation. Our findings indicate that there were varied responses to the COVID-19 pandemic among older, medically underserved people who smoked. Given the COVID-19 pandemic brought associated health, social, financial, and other stressors, these may have overpowered the health-related motivations to quit smoking due to reliance on tobacco as a maladaptive coping tool. Results may assist clinicians in addressing the role of tobacco use in response to highly stressful events.

## Supporting information

**S1 File. Interview guide.**
(DOCX)

## Author Contributions

**Conceptualization:** Ryan Coffman, Joanna Hart.

**Data curation:** Dorothy Sheu, Aerielle Belk, Jannie Kim.

**Formal analysis:** Tamar Klaiman, Aerielle Belk, Jasmine Silvestri, Jannie Kim, Joanna Hart.

**Funding acquisition:** Joanna Hart.

**Investigation:** Joanna Hart.

**Methodology:** Tamar Klaiman, Jasmine Silvestri.

**Project administration:** Dorothy Sheu, Aerielle Belk, Jannie Kim.

**Supervision:** Tamar Klaiman, Joanna Hart.

**Writing – original draft:** Tamar Klaiman.

**Writing – review & editing:** Tamar Klaiman, Nsenga Farrell, Dorothy Sheu, Aerielle Belk, Jasmine Silvestri, Ryan Coffman, Joanna Hart.

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
