## [Decision Letter · Decision Letter 0]

11 Jun 2024

PONE-D-24-07719Tobacco Use as Stress Relief During COVID-19PLOS ONE

Dear Dr. Klaiman,

Thank you for submitting your manuscript to PLOS ONE. After careful consideration, we feel that it has merit but does not fully meet PLOS ONE’s publication criteria as it currently stands. Therefore, we invite you to submit a revised version of the manuscript that addresses the points raised during the review process.

We look forward to receiving your revised manuscript.

Kind regards,

Enkeleint A. Mechili

Academic Editor

PLOS ONE

Journal Requirements:

2. Please describe in your methods section how capacity to provide consent was determined for the participants in this study. Please also state whether your ethics committee or IRB approved this consent procedure. If you did not assess capacity to consent please briefly outline why this was not necessary in this case.

 [PCORI ID: PCS-2018C1-11326 (JH)].  

4. Please expand the acronym “PCORI” (PCORI ID: PCS-2018C1-11326 (JH)) so that it states the name of your funders in full.

5. In the online submission form, you indicated that [The datasets generated and analyzed during the current study are not publicly available because informed consent stated that any data collected would not be used in future studies; however, de-identified data may be available from the corresponding author on reasonable request.]. 

6. We note that you have indicated that there are restrictions to data sharing for this study. PLOS only allows data to be available upon request if there are legal or ethical restrictions on sharing data publicly. For more information on unacceptable data access restrictions, please see http://journals.plos.org/plosone/s/data-availability#loc-unacceptable-data-access-restrictions. 

7. Please amend either the title on the online submission form (via Edit Submission) or the title in the manuscript so that they are identical.

8. Please amend either the abstract on the online submission form (via Edit Submission) or the abstract in the manuscript so that they are identical.

Reviewers' comments:

Reviewer's Responses to Questions

**Comments to the Author**

1. Is the manuscript technically sound, and do the data support the conclusions?

Reviewer #1: Yes

Reviewer #2: Yes

Reviewer #3: No

2. Has the statistical analysis been performed appropriately and rigorously? 

Reviewer #1: No

Reviewer #2: N/A

Reviewer #3: No

3. Have the authors made all data underlying the findings in their manuscript fully available?

Reviewer #1: Yes

Reviewer #2: Yes

Reviewer #3: No

4. Is the manuscript presented in an intelligible fashion and written in standard English?

Reviewer #1: Yes

Reviewer #2: Yes

Reviewer #3: Yes

5. Review Comments to the Author

Reviewer #1: The aim of the authors' study was to identify the impacts of the COVID-19 pandemic on tobacco use and preparedness for smoking cessation among individuals who smoke and are older and medically underserved. This work will assist clinicians understand the role of tobacco use in responding to high-stress events.

There are the following comments and suggestions:

1. The study was limited to Pennsylvania and was a cross-sectional survey. But the title of the article is "Tobacco Use as Stress Relief During COVID-19." So is this title not specific enough?

2. It is suggested to supplement the research methods: a) How to recruit people? b) Sample size; c) Interview topic list and whether everyone is interviewed according to the same interview topic list.

3. The author adopts a qualitative research method, and the data are presented by the interview focus and the discourse of the interviewee. Is it necessary to consider statistical analysis of the interview focus and interviewees' responses so as to better present the reliability of the results?

Reviewer #2: This was an interesting study with findings that are relevant to those within tobacco control. As this manuscript highlights, it would seem that there were missed opportunities during the pandemic to increase smoking cessation, particularly among populations like those identified in this paper who are already underserved and were/are at higher risk for COVID-19 infection and complications. I think I only have a couple of minor critiques for the authors to consider.

First, I would consider softening the causal sounding language in the Discussion section (e.g., "increases in stress...led to increased tobacco use") and any kind of reference to mechanism, given the study was both qualitative in nature and based on participant self-report. It may also be worth noting in the limitations that this study was purely observational.

Second, it may also be worth noting that not only were these observed coping patterns maladaptive, but I might emphasize that the behavior of smoking itself seems to put participants at increased risk of COVID-19 infection. For example, there is strong evidence that tobacco use increased risk of mortality and disease severity/progression among COVID-19 patients.

Reviewer #3: The sample size used by the author is too small to represent the entire population of smokers, leading to potential issues with the conclusions drawn from this data. Additionally, the data provided by the author mostly consists of subjective language descriptions from respondents, lacking scientific reliability.

6. PLOS authors have the option to publish the peer review history of their article (what does this mean?). If published, this will include your full peer review and any attached files.

Reviewer #1: No

Reviewer #2: No

Reviewer #3: No

---

## [Author Response · Author response to Decision Letter 0]

29 Jul 2024

Response to Reviewers

Thank you, the formatting has been updated to adhere to the style requirements. 

2. Please describe in your methods section how capacity to provide consent was determined for the participants in this study. Please also state whether your ethics committee or IRB approved this consent procedure. If you did not assess capacity to consent please briefly outline why this was not necessary in this case.

We have addressed this in the updated methods section. “Participants were given a full description of the study and agreed to participate after expressing understanding of the study procedures.”

 [PCORI ID: PCS-2018C1-11326 (JH)]. 

This has been added to the cover letter. 

4. Please expand the acronym “PCORI” (PCORI ID: PCS-2018C1-11326 (JH)) so that it states the name of your funders in full.

We have updated the cover letter to include this information. 

5. In the online submission form, you indicated that [The datasets generated and analyzed during the current study are not publicly available because informed consent stated that any data collected would not be used in future studies; however, de-identified data may be available from the corresponding author on reasonable request.]. 

Thank you. We contacted the University of Pennsylvania IRB who gave us guidance on this issue. We no longer have contact information for participants, and at the time they were consented, we were not required to include a future use statement in the consent form. We are not comfortable putting the data in a public repository without participant consent. 

6. We note that you have indicated that there are restrictions to data sharing for this study. PLOS only allows data to be available upon request if there are legal or ethical restrictions on sharing data publicly. For more information on unacceptable data access restrictions, please see http://journals.plos.org/plosone/s/data-availability#loc-unacceptable-data-access-restrictions. 

Thank you. As noted in the response to #6. We have contacted the University of Pennsylvania IRB to discuss this issue. We no longer have contact information for participants, and at the time they were consented, we were not required to include a future use statement in the consent form. We are not comfortable putting the data in a public repository without participant consent. 

7. Please amend either the title on the online submission form (via Edit Submission) or the title in the manuscript so that they are identical.

We have updated the title in the manuscript to reflect the title in the online submission form. 

8. Please amend either the abstract on the online submission form (via Edit Submission) or the abstract in the manuscript so that they are identical.

The abstract has been updated for consistency.

We have included a caption for our Supporting Information files..

Reviewers' comments:

Reviewer's Responses to Questions

Comments to the Author

5. Review Comments to the Author

Reviewer #1: The aim of the authors' study was to identify the impacts of the COVID-19 pandemic on tobacco use and preparedness for smoking cessation among individuals who smoke and are older and medically underserved. This work will assist clinicians understand the role of tobacco use in responding to high-stress events.

There are the following comments and suggestions:

1. The study was limited to Pennsylvania and was a cross-sectional survey. But the title of the article is "Tobacco Use as Stress Relief During COVID-19." So is this title not specific enough?

We have updated the title to “Use of Tobacco During COVID-19: A Qualitative Study among Medically Underserved Individuals.”

2. It is suggested to supplement the research methods: a) How to recruit people? b) Sample size; c) Interview topic list and whether everyone is interviewed according to the same interview topic list.

Thank you for your feedback. 

a) We have described our recruitment strategy in greater detail in the methods section.

b) We described the sample size and demographics in the results section.

c) We have noted that all respondents were asked questions about the key topics specified in the methods section.

3. The author adopts a qualitative research method, and the data are presented by the interview focus and the discourse of the interviewee. Is it necessary to consider statistical analysis of the interview focus and interviewees' responses so as to better present the reliability of the results?

Thank you for your suggestion. The purpose of the study was to learn about people’s behaviors and perspectives via in-depth interviews. The sample size is too small to conduct statistical analyses. We were interested in learning about peoples’ experiences directly from them, rather than testing hypotheses in a large sample. We have noted in the limitations section “Future work in this area should consider utilizing the concepts we identified as salient to develop a large-scale survey that would allow for statistical comparisons and analyses.”

Reviewer #2: This was an interesting study with findings that are relevant to those within tobacco control. As this manuscript highlights, it would seem that there were missed opportunities during the pandemic to increase smoking cessation, particularly among populations like those identified in this paper who are already underserved and were/are at higher risk for COVID-19 infection and complications. I think I only have a couple of minor critiques for the authors to consider.

First, I would consider softening the causal sounding language in the Discussion section (e.g., "increases in stress...led to increased tobacco use") and any kind of reference to mechanism, given the study was both qualitative in nature and based on participant self-report. It may also be worth noting in the limitations that this study was purely observational.

Thank you for your feedback. We have softened the language so as not to assume causal mechanisms. We have also noted in the limitations that “The study was purely observational, which limits our ability to assess causal mechanisms for increased tobacco use.”

Second, it may also be worth noting that not only were these observed coping patterns maladaptive, but I might emphasize that the behavior of smoking itself seems to put participants at increased risk of COVID-19 infection. For example, there is strong evidence that tobacco use increased risk of mortality and disease severity/progression among COVID-19 patients.

We have included that patients who smoke had worse outcomes from COVID-19 with a corresponding citation. 

Reviewer #3: The sample size used by the author is too small to represent the entire population of smokers, leading to potential issues with the conclusions drawn from this data. Additionally, the data provided by the author mostly consists of subjective language descriptions from respondents, lacking scientific reliability.

This is a qualitative study using reliable scientific qualitative methods. We were interested in learning about peoples’ experiences directly from them, rather than testing hypotheses in a large sample.

---

## [Editor Report · Decision Letter 1]

5 Aug 2024

Use of Tobacco During COVID-19: A Qualitative Study among Medically Underserved Individuals

PONE-D-24-07719R1

Dear Dr. Klaiman,

We’re pleased to inform you that your manuscript has been judged scientifically suitable for publication and will be formally accepted for publication once it meets all outstanding technical requirements.

Kind regards,

Enkeleint A. Mechili

Academic Editor

PLOS ONE
---

## [Editor Report · Acceptance letter]

8 Aug 2024

PONE-D-24-07719R1 

PLOS ONE

Dear Dr. Klaiman, 

I'm pleased to inform you that your manuscript has been deemed suitable for publication in PLOS ONE. Congratulations! Your manuscript is now being handed over to our production team.

Kind regards, 

on behalf of

Prof. Assoc. Dr. Enkeleint A. Mechili 

Academic Editor

PLOS ONE